# A mixed methods study of community-based health insurance enrollment trends and underlying challenges in two districts of northeast Ethiopia: A proxy for its sustainability

**Mohammed Hussien**[1]*, **Muluken Azage**[2], **Negalign Berhanu Bayou**[3]

1 Department of Health Systems Management and Health Economics, School of Public Health, College of Medicine and Health Sciences, Bahir Dar University, Bahir Dar, Ethiopia, 2 Department of Environmental Health, School of Public Health, College of Medicine and Health Sciences, Bahir Dar University, Bahir Dar, Ethiopia, 3 Department of Health Policy and Management, Faculty of Public Health, Institute of Health, Jimma University, Jimma, Ethiopia

* muhamedun@gmail.com

**Data Availability Statement:** All relevant data are within the paper.

## Abstract

### Background

The term "community-based health insurance" refers to a broad range of nonprofit, prepaid health financing models designed to meet the health financing needs of disadvantaged populations, particularly those in the rural and informal sectors. Due to their voluntary nature, such initiatives suffer from persistently low coverage in low- and middle-income countries. In Ethiopia, the schemes' membership growth has not been well investigated so far. This study sought to examine the scheme's enrollment trend over a five-year period, and to explore the various challenges that underpin membership growth from the perspectives of various key stakeholders.

### Methods

The study employed a mixed methods case study in two purposively selected districts of northeast Ethiopia: Tehulederie and Kallu. By reviewing the databases of health insurance schemes, quantitative data were collected retrospectively from 2017 to 2021 to examine enrollment trends. Trends for each performance indicator were analyzed descriptively for the period under study. Face-to-face interviews were conducted with nine community members and 19 key informants. Study participants were purposely selected using the maximum variation technique. Interviews were audio recorded, transcribed verbatim, and translated into English. Thematic analysis was employed with both deductive and inductive coding approaches.

### Results

Over the course of the study period, enrollment in the scheme at both districts exhibited non-linear trends with both positive and negative growth rates being identified. Overall, the

**Funding:** The authors received no specific funding for this work.

**Competing interests:** The authors have declared that no competing interests exist.

scheme in Tehulederie has a relatively higher population coverage and better membership retention, which could be due to the strong foundation laid by a rigorous public awareness campaign and technical support during the pilot phase. The challenges contributing to the observed level of performance have been summarized under four main themes that include quality of health care, claims reimbursement for insurance holders, governance practices, and community awareness and acceptability.

## Conclusions

The scheme experienced negative growth ratios in both districts, indicating that it is not functionally viable. It will fail to meet its mission unless relevant stakeholders at all levels of government demonstrate political will and commitment to its implementation, as well as advocate for the community. Interventions should target on the highlighted challenges in order to boost membership growth and ensure the scheme's viability.

## Introduction

Out-of-pocket (OOP) payment is the main source of health care financing in low- and middle-income countries [1, 2]. Millions of people were unable to seek health care because it must be paid for at the time of use [3, 4]. It is also a signal that people are more likely to face financial hardships as a result of receiving health care [5]. To achieve universal health coverage (UHC), a strong commitment is required to shift from an OOP payment model to a prepaid, pooling strategy that expands access to essential care while spreading the financial risks of illnesses across the population [3, 4]. In this regard, community-based health insurance (CBHI) has got a prominent place in low- and middle-income countries. CBHI is a generic term that covers a variety of nonprofit health financing schemes initiated by communities, providers, enterprises, and others to meet the health care financing needs of disadvantaged populations especially in the rural and informal sectors [6, 7]. In theory, all CBHI schemes share common characteristics including solidarity, where risk sharing is as inclusive as possible and membership premiums are independent of individual health risks; community-based social dynamics, where the schemes are organized by and for individuals who have predominantly low income, earning a subsistence from the informal sector, or are socially excluded and share common characteristics; participatory decision-making where members are actively involved in driving the design and management of the scheme; nonprofit character; and voluntary affiliation [8–10].

CBHI is appropriate for subsistence farmers and workers in the informal sector, because they are typically not covered by payroll-based social health insurance programs in many countries [9]. It has been adopted in a growing number of sub-Saharan African countries, including Ethiopia, as part of the effort to meet the health care needs of informally employed and low-income people, which make up the majority of the population [1, 8]. The sustainability of a voluntary CBHI scheme relies to a greater extent on the ability of implementers to attract and retain members [11]. The key performance indicators for measuring membership development are growth ratio, coverage ratio, and renewal ratio [11, 12].

A large membership base enhances economies of scale, risk pooling, and reduced vulnerability to unforeseen events. It allows for the retention of healthy members while avoiding adverse selection, resulting in higher revenue, lower marginal costs, and lower health care spending [11]. In the event of decreasing coverage, members who remain in the scheme are more likely to be chronically ill and in high-risk age groups. This leads to higher claims costs, jeopardizing the schemes' financial viability [11–13].

With the exception of a few success stories, CBHI initiatives in low- and middle-income countries are plagued by persistently low coverage due to their voluntary orientation [8, 14]. A review of studies in four sub-Saharan African countries showed that the existence of a large informal sector whose members are mostly uninsured, and a high dropout rate, were among the main challenges facing CBHI schemes to sustain and achieve the goal of UHC [15]. Earlier studies have identified a variety of challenges that contribute to low membership coverage. The most prominent challenge is poor quality of health care [16–20]. Other plausible reasons include the lack of awareness of the risk-sharing principle and the benefits of insurance plans [21–24], claims rejections [25], and members' lack of trust in the scheme's integrity [24, 26].

It is essential to generate empirical evidence on the success of CBHI schemes for overcoming existing barriers as well as initiating structural and design changes [27]. In Ethiopia, several studies have been conducted on enrollment [28–33] and membership renewal decisions [34–36]. However, none of these explored the different issues related to membership development from the perspectives of various stakeholders. A qualitative study examined the barriers and facilitators of membership growth; however, it was solely based on community members' perspectives [24]. Therefore, the current study sought to examine the scheme's enrollment trend over a five-year period from 2017 to 2021, and to explore the challenges that underpin membership growth from the perspectives of various key stakeholders using mixed methods research.

Based on lessons learned from previous implementations, Ethiopia's current health care finance strategy aims to establish a unified pool system at a national level that allows cross-subsidy between high-risk and low-risk areas [37]. The findings of this study will be valuable to policymakers and other relevant stakeholders to overcome implementation challenges and develop membership attraction and retention strategies in an effort to establish the proposed higher-level pools.

## A brief overview of CBHI in Ethiopia

Since July 2011, Ethiopia is implementing a CBHI scheme to meet the health care needs of rural households that constitute an estimated 85% of the country's population. The scheme was launched in 13 pilot districts in four regional states as part of the health care financing reform aimed at reaching the goal of UHC [25]. Based on the evaluation findings of the pilot initiative, the scheme was first expanded to 161 districts in July 2013, and then to 827 districts as of July 2020 with a total enrollment coverage of 50% [38].

The government is in charge of CBHI, with an active participation of the community in its design and implementation. Membership is voluntary and must be renewed annually. To reduce adverse selection, the membership unit was set at a household level [25, 39].

The scheme's primary sources of revenue are government subsidies, premiums, and registration fees. The federal government provides 25% of annual enrollment contributions to the scheme as a general subsidy. In addition, regional and district governments provide a targeted subsidy to cover the costs of fee waivers for 10% of the target population who are indigent. Premiums are set at the household level based on core family members (a mother, father, and their children under the age of 18) and additional annual premiums must be paid for each non-core family member. Regional states have the authority to update the premium based on local circumstances [25, 39]. For instance, the initial yearly premium in the Amhara Regional State, where this study is conducted, was 8.34 USD regardless of family size [25]. Eventually it was changed to varying levels of contributions dependent on family size. At the time of the study, the annual premium in rural areas ranged from 8.89 USD to 12.19 USD with 2.54 USD for each non-core family member [40].

The benefit packages cover all outpatient and inpatient services at health centers and hospitals within Ethiopia, with the exception of cosmetic treatments, organ transplants, chronic renal dialysis, treatment for exempted services, and non-generic medicines [25, 39, 40]. Specific to the Amhara Regional State, members must follow the referral path in order to obtain free health care via the CBHI. The scheme will not cover treatment costs if individuals bypass health centers and seek care from hospitals without a referral letter. The service providers for scheme cardholders are public health facilities. All health services and medicines covered by the benefit packages are provided to scheme members free of charge at CBHI affiliated health facilities. Scheme members can only claim reimbursements for expenses made at private institutions for services or medicines that are not available in contracted health facilities as long as they follow the formal procedure and submit the necessary paperwork [40].

## Materials and methods

### Study setting

The study took place in two rural districts of northeast Ethiopia: Tehulederie and Kallu. Tehulederie is divided into 20 rural and seven urban *Kebeles* (subdistricts) with a population of 145,625, of which 87.5% are residing in rural areas. The district has five health centers and one primary hospital. Kallu is divided into 36 rural and four urban *Kebeles*, with nine health centers. It is the most populous district in the zone, with a population of 234,624, with 89.11% residing in rural areas [41]. For the vast majority of the population in the study area, agriculture is the primary source of income.

Tehulederies was one of the CBHI pilot districts in Ethiopia, which began implementing the initiative in July 2011. Two years later, the scheme was launched in Kallu, in July 2013. The district-level scheme is part of the health sector and is governed by the health insurance board. The board signs a contract with public health facilities annually, and reimbursements are made at the end of every three months based on a fee-for-service payment approach. Health facilities must receive payments within two weeks after filing their claims. The scheme conducts a medical audit before reimbursing them, and it is likely that claims will be deducted based on the audit findings. The scheme also reimburses insurance holders for OOP expenses made in private institutions as long as they follow the right procedure. At a *Kebele* level, the key players for membership enrollment, renewal, and premium collection are *Kebele* leaders and health extension workers (HEWs). One of the HEWs' responsibilities as community health workers is to persuade people for enrollment.

### Study design

The study employed a mixed methods case study with both quantitative and qualitative data collected simultaneously. Mixed methods research is an approach to inquiry involving collecting both quantitative and qualitative data, integrating the two forms of data, and organizes these procedures into specific research designs [42]. We applied a qualitatively driven, concurrent nested design in which the quantitative component was embedded within the primary qualitative study to answer a complementary question [43]. The aim of the quantitative part was to assess the performance of the CBHI scheme in terms of membership development using key measurement indicators while the qualitative part was intended to explore the underlying challenges that impede membership development efforts. Thus, the driving motive for combining the two approaches in this study is the belief that both kinds of research have value, that they generate different but complementary data which offer a holistic view of the scheme's membership development.

Results from both the quantitative and qualitative components are presented separately, and the two components are integrated at data interpretation stage. The findings from the qualitative and quantitative components are discussed, and connections are made between the various challenges explored through the qualitative interviews and the level of performance observed in terms of membership development.

## Participant selection

The study approach considered each of the two districts as a separate case study of the scheme's performance. We used purposive sampling to select the two study districts. Tehulederie was the sole early adopter of the scheme in the zone, serving as a pilot district, hence selected as an outlier case. The second case (Kallu), was selected as a typical case since it is the zone's largest district that shares a variety of geographical features with other districts. Outlier sampling or deviant case sampling involves selecting cases that are information rich, because they are unusual or special in some way, while typical case sampling involves selecting and studying cases that are average to understand, illustrate, and highlight what is typical and normal [44].

Qualitative interview participants in both study sites were purposely selected using the maximum variation technique in order to gain insight from a diverse range of viewpoints and to chronicle important shared experiences that cut across the various stakeholders participating in the CBHI scheme implementation [44]. Key informants were selected among stakeholders based on their active participation in the scheme's implementation and their ability to provide a wealth of data. A total of 19 key informants (eight in Tehulederie and 11 in Kallu) were recruited by considering the different sectors, that included two scheme personnel, one district health officer, four health center directors, five health care providers, three *Kebele* leaders and four HEWs. Two of the key informants were coworkers of the lead investigator as health care providers in one of the study districts. Nine community members (five in Tehulederie and four in Kallu) were selected by HEWs for in-depth interviews based on their insurance status (current and previous members), and their ability to provide useful information. HEWs are familiar with the population in rural *kebeles* because their main responsibility is to provide home-based health services and mobilize the community for health insurance. The final sample size at each study district was determined based on data saturation, with no new information emerging from participants [45]. Individuals with similar characteristics to the formal interviewees at district health offices, health facilities and in the community were also invited for informal interviews based on the relevance of the information they provided during informal interactions.

## Data collection

The data was collected between February 8 and May 2, 2021. Quantitative data was gathered by reviewing the databases of the two CBHI schemes retrospectively using checklists developed based on key performance indicators. Data on the eligible target population, the number of new enrollees, and expected as well as actual renewals were collected for each enrollment period under consideration. Although we intended to examine all years of the scheme's implementation (10 years in Tehulederie since 2011 and seven years in Kallu since 2013), we were only able to get complete data from 2017 to 2021.

Qualitative data were collected using key informant interviews (KII), in-depth interviews (IDI), and informal field interviews (IFI) [44]. The IDI was conducted with current and previous members of the CBHI to explore their views and experiences concerning health care quality, community willingness to participate in the scheme, claim benefits, and scheme services.

The IDIs were conducted at health posts (HEW's office). The KIIs intend to explore the views of different stakeholders regarding community understanding and acceptability of the CBHI, health care quality, and claims management. Key informants were interviewed at their offices based on a pre-specified schedule. Informal interviews were also made during our visits to the district health offices, health facilities and households to capture important information that could triangulate with the formal interviews. Informal interviews are those that are conducted either with a single participant in natural conversation or with some small group of people, asking normal, conversational questions during periods of informal interaction [44].

All the formal interviews were conducted face-to-face by the lead investigator in convenient locations using an interview guide that was designed to elicit the participants' views via open-ended questions which were further probed to trigger more discussions. Interviews were conducted in Amharic, the local language, and lasted between 10 and 40 minutes. All the formal conversations were audio recorded using a digital voice recorder with the permission of the participants. Field notes were taken during informal interviews. Every person we approached agreed to take part in the study.

## Data analysis

Quantitative data obtained through document review were analyzed using the performance indicators selected for this study, which include membership growth ratio, coverage ratio, and renewal ratio. The coverage ratio is the proportion of the target population who are insured, whereas the renewal ratio is the proportion of insured households who renewed their subscription during the current period among those eligible to renew. The growth ratio is a combination of coverage and renewal ratios that measures the proportion of the number of insured people who have increased or decreased over time [11, 12]. The target population in this study refers to the estimated total population in the study area who are eligible for CBHI membership, which include farmers and those working in the informal sector. Those illegible for renewal refer the number of potential renewals (the number of clients that could have renewed their coverage) [12]. The trends of each indicator over the study period were analyzed using Microsoft Excel.

Audio records from the qualitative interviews were transcribed verbatim, and then translated into English. Field notes were made part of the transcription. Thematic analysis was done based on the finalized translated data. The Atlas.ti 9 software package was used to facilitate the coding process. To begin, all the interview transcripts were read and reread to have a thorough understanding of the data set. Both deductive and inductive approaches were used in the coding process. The deductive approach was used as the starting point by defining preliminary themes and sub-themes based on the research questions and a review of existing literature. Themes that emerged during the coding process were added inductively, rather than trying to fit them into a preexisting coding framework. The lead investigator generated initial coding schemes and categorized the codes into subthemes and themes independently. The preliminary results were reviewed by the coauthors and agreements were reached through ongoing discussions.

We used different approaches to enhance trustworthiness of the data, including data triangulation, thick description, and reflexivity. The different data collection methods and data sources (method and source triangulation) could increase the credibility of the findings [46]. To foster reflexivity, the principal investigator kept a reflexive journal to record expectations, feelings, observations, interview experiences, choices available, and decisions made throughout the research process [42]. A rich, thick description of the research setting, process, and findings was made to enhance its understanding and transferability. We provide a detailed account

of the findings, which was backed up by adequate evidence in the form of quotes from participant interviews [45].

## Ethical considerations

Ethical approval was obtained from the Institutional Review Board of the College of Medicine and Health Science, Bahir Dar University, before commencing the study. A support letter was communicated to the district health offices to gain entry permission into the study areas. Each of the study participants gave their verbal informed consent. Consent was obtained on the use of a voice recorder during interviews. Confidentiality was assured through collecting anonymous information and informing the participants that personal identifiers would not be revealed to a third party.

## Results

### Enrollment trends in the scheme

Over the period under study, the number of insured households ranged between 11,535 and 14,188 in Tehulederie and 12,175 to 27,859 in Kallu, with an estimated target population of 22,216 and 45, 876, respectively. The coverage ratio in the 2021 enrollment period was 65.5% in Tehulederie and 58.5% in Kallu.

The five-year enrollment trend of the scheme in both districts is displayed under Table 1. In Tehulederie district health insurance scheme, population coverage dropped from 65.7% in 2017 to 62.9% in 2018 and to 51.3% in 2019. It increased back to 60.2% in 2020 and to 65.5% in 2021. In 2018 and 2019, a negative growth ratio of 1.3% and 17.6% was observed, while a positive growth ratio of 18.3% and 3.7% was recorded in 2020 and 2021, respectively. The renewal ratio ranged from 76.7% in 2019 to 96.6% in the 2021 renewal period.

The population coverage in Kallu district health insurance scheme was 27.2 in 2017. It increased to 57.9% in 2018 and dropped back to 50.3% in 2019. A 60% and 58.5% coverage were reached in 2020 and 2021, respectively. The growth ratio did not show a linear increase or decrease during the period under study. A positive growth ratio of 116.3% and 19.8% was observed in 2018 and 2020, while a negative growth ratio of 11.7% and 2.5% was achieved in 2019 and 2021, respectively. The renewal ratio ranged from 37% in 2017 to 68.3% in the 2018 registration period. Fig 1 depicts the enrolment trend of the two schemes over a five-year period.

### Challenges to membership development

While each case study site is assessed as a unique implementation of the CBHI, integrating findings from the two case studies allows for the identification of key common themes. Four main themes were identified, which are quality of health care, claims reimbursement for

**Table 1. Coverage, renewal and growth ratios of CBHI membership (2017–2021) at two case study districts of northeast Ethiopia.**

|  | Tehulederie | | | Kallu | | |
|---|---|---|---|---|---|---|
| Year | Coverage ratio (%) | Renewal ratio (%) | Growth ratio (%) | Coverage ratio (%) | Renewal ratio (%) | Growth ratio (%) |
| 2017 | 65.7 | 89.9 | - | 27.2 | 37.0 | - |
| 2018 | 62.9 | 94.3 | -1.3 | 57.9 | 68.3 | 116.3 |
| 2019 | 51.3 | 76.7 | -17.6 | 50.3 | 56.8 | -11.7 |
| 2020 | 60.2 | 80.5 | 18.3 | 60.0 | 63.6 | 19.8 |
| 2021 | 65.5 | 96.6 | 3.7 | 58.5 | 62.6 | -2.5 |

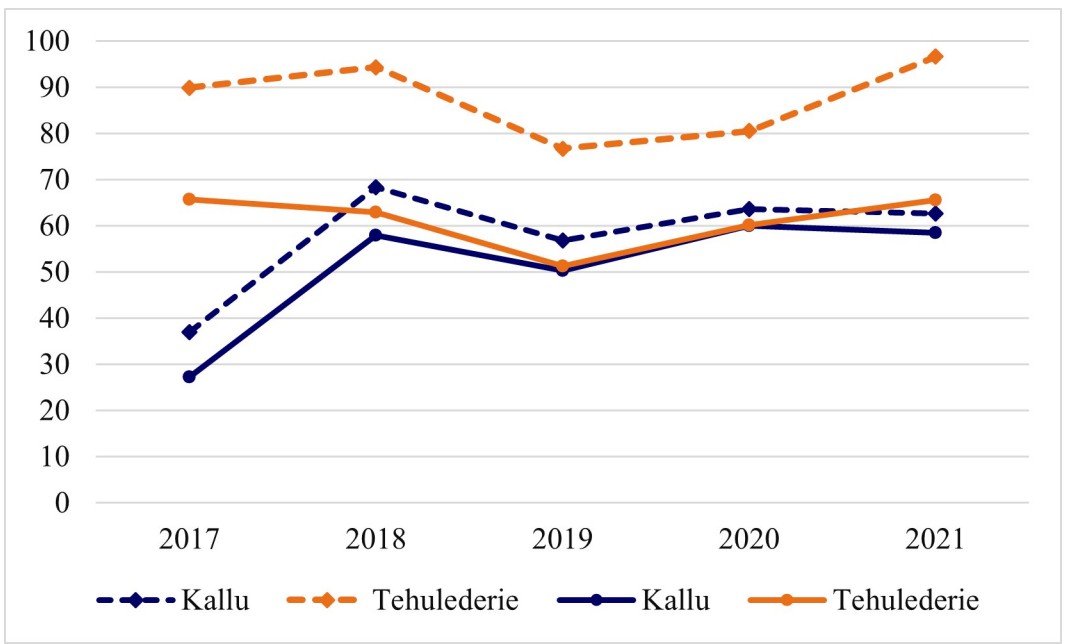

**Fig 1. Trends in community-based health insurance enrollment in two case study districts of northeast Ethiopia between 2017 to 2021.** Note: Solid lines represent coverage ratios while broken lines represent renewal ratios, both in percentages.

insurance holders, governance practices, and community awareness and acceptability of the CBHI (Fig 2). Similar sub-themes are categorized and described under each main theme.

## Quality of health care

*Availability and perceived quality of medicines*. Unavailability of medicines was the most frequently discussed issue in all interviews. Due to the unavailability of medicines in contracted health centers and hospitals, insured patients were usually given prescriptions to buy from private pharmacies, and forced to make OOP spending or forgo treatment if they were not able to afford its cost. In addition to health service users, health care providers and health center directors from both districts were well aware of the depth of this problem. The two main reasons for medicine stock outs at health centers were a lack of budget and limited capacity of the government's pharmaceutical supply agency. Especially, key informants emphasized the pharmaceutical supply agency's limited capacity to meet the demands of health facilities in its catchment area. Because the government's supplier agency frequently runs out of medical supplies, health facilities have been compelled to procure from private vendors, which requires a lengthy process and higher price markups.

> "*The biggest problem now is that there is a shortage of medicines in all health centers that provide services to insured patients, and most patients have to purchase their medications from private pharmacies at a higher cost.*" (KII-5, *health center director*)

In addition to the lack of medicine availability, IDI participants believe that medicines provided at health centers are of poor quality. The majority of insured patients fail to take their prescribed medications as directed, which may be due to a lack of faith in their quality. Health care providers disagreed that this was the community's judgment based on their preferences, expectations, and level of awareness regarding medicines.

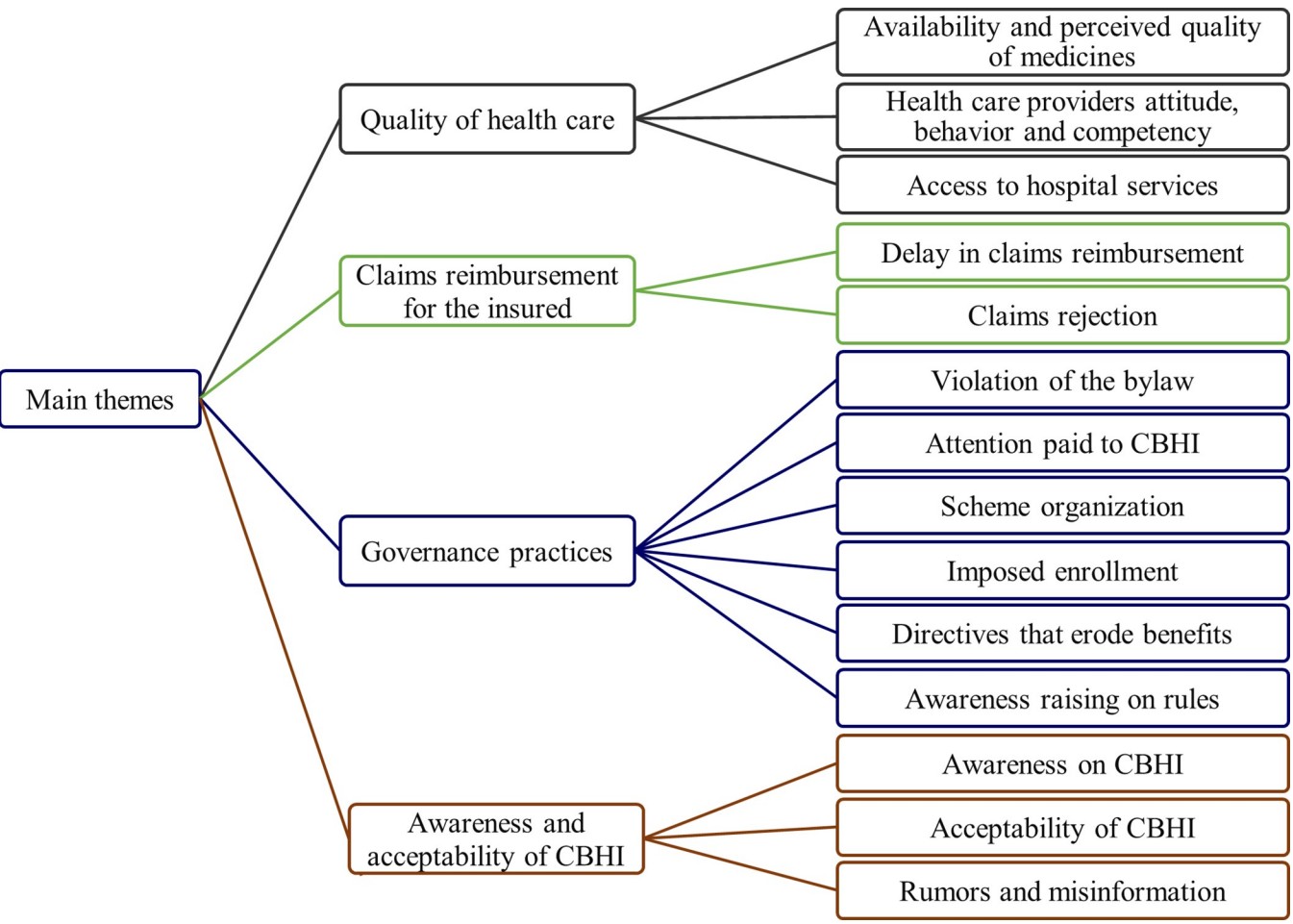

**Fig 2. Main themes and sub-themes from interviews with CBHI members and key informants at two case study districts of northeast Ethiopia.**

In general, the community prefers injectables over pills, and compares medicines prescribed in public health facilities to those prescribed in private clinics. In public facilities, medicines are prescribed based on the standard treatment guidelines, whereas in private clinics, fast-acting injectables, and new generation medicines with brand a form are prescribed for most patients. However, the wider public is unaware of the benefits and risks of these approaches. Furthermore, health facilities are reimbursed for services rendered in accordance with standard treatment guidelines.

*"I recover from my illness quickly if I get injectables. I don't like swallowing medicines; it does not cure me. Why should we take their rotten medicines?" (IFI, current member)*

*Health care providers attitude, behavior and competency.* One of the major concerns in the service delivery process of contracted health facilities is perceived discriminatory treatment of insured patients in favor of paying patients. Health service users stated that health care providers would negatively judge, label, verbally abuse, humiliate or belittle them owing to their insurance coverage. Some health care providers have also been accused of lacking good human relations, behaving rudely, speaking to patients in an unprofessional manner, not being caring and empathetic, and neglecting to assist them.

"*My mother became ill and I took her to a health center, they (the health care providers) gave priority to paying patients while we waited for care. The one who has money is getting ahead, and the one with health insurance is waiting long.*" (IDI-5, previous member)

"The problem now is that health workers prioritize paying patients and delay insured patients despite they get very sick." (KII-18, Kebele leader)

Health care providers admitted that they may not demonstrate excellent interpersonal communication or spend enough time examining and discussing patients' health concerns due to increased workload associated with insurance coverage. They disagreed, however, on the assertion that insured patients are treated differently in favor of paying patients. According to them, insured individuals associate everything that happens in health facilities with their insurance status. They believe that health care providers have abandoned them and that paying patients are given higher priority, based on the notion that insured clients attend health facilities for every minor ailment because of free health care. This is perceived discrimination, most of which did not really happen, and such issues were common during the early stages of CBHI, when health care providers were unfamiliar with the program, as most key informants stated.

"*My wife got sick and we went to a health center for consultation. They told us to go to a referral hospital without a referral letter. I told them that if they did not write the referral letter, the hospital would refuse us. They told me something wrong, they insulted me. The health center is just a name, it is . . .. It is not considered as if it is existing. Not complete treatment, let alone they do not speak a good language.*" (IDI-5, previous member)

"*Patients expect you to examine them thoroughly and do laboratory tests. This is the only way to make them happy. However, due to the high volume of patients, we are unable to fully deliver these services. . . . some clients allege, 'They dispense me medications chopped into parts because I am a health insurance beneficiary'.*" (KII-9, health care provider)

Furthermore, some participants believed that health care providers in health centers lacked the skills and experience to manage their conditions. They claimed that the prescribed medicines do not much the disease, providers treat patients without physical examination or laboratory tests and those working in rural areas are inexperienced. As a result, most insurance holders distrust health care providers and believe they are not there to assist them. As stated by health care providers, insured patients are not willing to accept professional advice and suggestions because they believe that providers are in opposition to them.

"*The health center is structurally sound, but it is devoid of competent personnel. It's possible to conclude that it's empty.*" (IDI-6, current member).

"*Insurance subscribers are frequently blamed by health care providers. As a result, we assume they have an unfavorable attitude toward us and treat us badly.*" (IFI, current member).

"*When you advise them, they will not believe you.*" (KII-8, health care provider)

*Access to hospital services.* Health service users expressed a variety of concerns about access to health services at the referral hospital. These include long waiting times, being made to pay in the event of an emergency if they don't have a referral letter from health centers, long appointments for non-emergency conditions, and service denial due to payment and contract issues. Despite the fact that CBHI subscribers are entitled to hospital services without a referral letter from health centers in an emergency, they are either forced to pay the treatment fee or

denied the service if they are unable to pay. Some people prefer private hospitals because they want immediate care and do not want to wait for extended appointments, but they are subjected to high treatment costs. Patients are also required to buy medicines in private pharmacies despite the fact that they are available in the hospital pharmacy, according to some study participants. In addition, one key informant revealed that insured individuals who seek exempted services at the hospital are ordered to buy medications from private pharmacies on purpose.

> "*I had a sudden illness before. I get better off with IV drugs in private hospitals. If I go for medical treatment using a health insurance card, I would die because they do not treat me right away.*" *(IDI-2, current member)*

> "*Something is worrying you; the scheme is helpful for appointment-based treatments, but it is useless in emergency situations. Many people believe that in the event of an emergency, treatment at a private hospital is mandatory.*" *(KII-18, Kebele leader)*

> "*Insured patients have been denied health services for more than three or four months in the hospital due to delays in claims reimbursements.*" *(KII-14, HEW)*

## Claims reimbursement for insurance holders

The issue of claims reimbursement appears to be another major impediment to CBHI implementation in both districts. Almost all respondents indicated that a delay in claims reimbursement was one of the key problems experienced by insured households. In Tehulederie, clients are frequently scheduled for several appointments and repeatedly travel to the health insurance office. They must also wait in long queues at the scheme's office to have their claims processed. To this end, they incur higher transportation costs and waste their time. Members opt not to submit smaller claims because of the extra costs of claims processing. This is not an issue in Kallu district, where reimbursements are made by HEWs at each *Kebele*. HEWs collect medical bills from scheme participants, process payments at the scheme office and disburse the money to participants. Although this reimbursement approach worked well in Tehulederie district, later HEWs abandoned it because of the increased workload and their fear of being held accountable, arguing that they had no obligation.

In addition, claims settlement delays are likely to result in the loss of documents and the closure of the fiscal year, which leads to claim rejection. According to scheme personnel, the delay was mainly caused by a budget deficit and a medical auditing process that must be completed before claims can be reimbursed.

> "*There are situations when reimbursing claims is too late. There are several scheme members who struggle for six, seven, or eight months and more until the end of the year, after which it is impossible to receive their claim. I don't meant to say that they have been completely denied, but people are getting increasingly frustrated. . .*" *(IDI-1, current member)*

> "*We are advised to travel to the CBHI office to collect pocket money replacements. However, we are incurring additional transportation costs, even in exchange for small reimbursements. As a result, many people choose not to request claims less than 100 ETB [Ethiopian Birr].*" *(IFI, current member)*

According to the majority of the in-depth interview participants, another concern they faced regarding pocket money reimbursement was claims rejection. They expressed their dissatisfaction by alleging that, despite submitting all required evidence, the scheme rejected their

appeals for a variety of reasons. Scheme personnel also acknowledged that claims rejection is common for a number of reasons, most of which stem from clients' misunderstanding of the kind of paperwork they should submit with their claims. They also disclosed that members of the scheme were unwilling to accept the rejection decision, regardless of the cause.

Presenting with illegal bills from private pharmacies, presenting bills from private pharmacies without attaching prescription papers from contracted facilities, mismatch between medications prescribed by physicians and those dispensed by private pharmacies, the prescription paper lacking the required signature and stamp, submitting claims after a deadline has passed, receiving treatment from a health facility that did not make a contract agreement with the scheme, receiving treatment in hospitals without being referred by health centers, and loss of submitted documents were some of the main reasons for claims rejection. One reason for the loss of submitted documents, according to a health care provider, is that documents presented by clients are likely to be disguised on purpose, especially for larger claims, because in some cases, there are players between the scheme and its members.

> "*Last year, the referral hospital refused to sign a contract agreement with the scheme. Insurance members were unaware that the deal with the hospital had not been reached. They sought medical treatment, paid the bill, and then requested reimbursement, but their claims were rejected.*" (KII-13, HEW)

> "*Every month, I have a follow-up appointment for hypertension treatment in a hospital. When the hospital's pharmacy is out of medicine, I buy it from private pharmacies. Surprisingly, I've never been reimbursed for my pocket money. I gathered all past bills and attempted to get reimbursed, but they said it was too late and rejected my request.*" (IFI, current member)

> "*We conducted a medical audit and rejected a lot of claims due to submission of illegal documents.*" (KII-16, CBHI personnel)

## Governance practices

One of the scheme's governance challenges is the lack of attention paid to its execution by higher-level authorities. Health insurance is only taken into consideration by higher authorities once a year during membership registration, after which no one is interested in supporting the scheme. They are unwilling to work together to solve the many problems that arise during implementation. According to some key informants, their primary concentration is on increasing the number of subscribers, rather than striving to serve the needs of the community. Due to a lack of technical and material assistance, scheme administrators were unable to undertake proper medical audits. Furthermore, the workload associated with the implementation of health insurance is not taken into account when it comes to equipping health facilities with the necessary infrastructure and personnel.

> "*The government did not pay attention to the implementation of health insurance. Although it is a useful institution, it has been neglected. Regional and zonal bodies appear only during the renewal period. For them [higher officials], health insurance is a campaign that only takes place once a year.*" (KI-15, CBHI personnel)

District health insurance schemes have been organized and are working within the district health office; however, key informants believe that the current structure poses challenges to the scheme's performance. They claimed that a single sector should not be both a provider and a buyer of health care, because it creates accountability concerns. They proposed that health

insurance be organized as a separate autonomous sector with clearly defined roles and responsibilities, as well as a mechanism that allows active engagement of other sectors in the district. The service seller would be accountable for the quality of health care provided to the insured, while the service buyer would be accountable for claim reimbursements in their interaction.

> "*Do you think the health office would sue the health center if scheme members are not treated properly? Would the health center sue the health office if the service charge is not reimbursed timely? Is it possible for one pocket to sue the other? This is something unscientific. This is what makes the scheme ineffective.*" (KII-1, district health officer)

According to the participants, another governance concern is that the rules governing the CBHI are being violated, as a result of which members are losing their benefits. The provision of health care to members and their families begins when the coverage ratio at the district level exceeds 60% of the target population, according to the bylaw. Participants in Kallu district complained that this rule is enforced at a *Kebele* level. As a result, certain *Kebeles* that did not achieve this objective were denied health care coverage for more than two months, including households who paid the premium. Some even object to the rule, arguing that once a person pays the premium, he or she has the right to receive the benefit package immediately.

> "*This year, certain segments of society have yet to benefit from health insurance. This is due to the fact that the Kebeles' membership coverage is less than 60%. As a result, persons with chronic health conditions would pay for the health care charge for a month or two without insurance coverage, even if they renew early.*" (KII-9, health care provider)

> "*After paying the premium, we must wait two months before receiving health care until everyone has been enrolled. Our family probably would get sick within the two month timeframe. Meanwhile, we are going to spend some money. In addition, I may not require treatment then after.*" (IDI-8, previous member)

At the time of this study, the release of a new guideline on the implementation of claims reimbursement for the insured was a hot topic. The regional health bureau released a directive stating that claims reimbursements should be made at the price set by public health facilities, which is far lower than the price in private institutions. Stakeholders participating in the implementation of the CBHI have voiced worries about the impact of the directive on future membership growth efforts as well as the risk of undermining benefits for scheme members. Let alone this directive, the community was not satisfied with the existing reimbursement system. It would be another major impediment to the community's participation in the scheme.

> "*If the current law about pocket money reimbursement remains in place, no one wants to join the health insurance plan. For example, one member had treatment that cost 5,000 ETB, but he was only paid 1200 ETB under the new rule. 'What is the significance of health insurance?, what happens if I drop out of the scheme and have to pay for my own treatment?' he said.*" (KII-15, CBHI personnel)

> "*..., why do people suffer? If this is the case, we do not need to be covered by health insurance. When you're sick, it's better to go to a private facility. Either the government must provide all services through its own health facilities or private organizations must limit the pricing of medicines. Otherwise, we will not find anyone to register in the future.*" (KII-18, Kebele leader)

Insured households are also unable to receive the full range of CBHI benefits due to a lack of awareness of the procedure they should follow. Community members who took part in the interviews claimed that they were not aware of important directives when it came to their benefits. This is the main reason for the high claims rejection ratio that has been reported.

*"The goal of health insurance is to be profitable. The community was unaware that the scheme would reimburse for pocket money. The insurer does not want the public to know. The community became aware of this regulation after a long period." (IFI, current member)*

Members of the community complain that, despite participation being voluntary, the *Kebele* leaders are forcing them to enroll and renew their membership. *Kebele* leaders, HEWs and scheme administrators have also admitted that there is some form of "positive enforcement" for membership enrollment and renewal. Some argue that CBHI is politicized, because refusing to enroll is often seen as a sign of opposition to government policies. Enforced enrollment erodes the community's sense of ownership and leads people to develop a negative attitude toward health insurance.

*"We are being forced to pay the health insurance membership fee by Kebele leaders. They will refuse to receive labor tax if we do not pay our health insurance contributions in advance, and if we refuse, we will be jailed. If they force the community, it appears to them that they are being forced for the government's advantage rather than their own." (IFI, current member)*

*". . ., when you see the majority, they are forced by the Kebele leaders, and we are going home to home and ask them to renew their membership." (KII-14, HEW)*

*"People should join health insurance voluntarily. If they are forced to join the scheme, and they become upset over something in the health facility, they will complain that something big has happened and are prone to quite their membership (KII-9, health care provider).*

According to a *Kebele* leader from a high-performing area, *Kebele* administrators' commitment is vital to membership development. He stated that his *Kebele* was able to achieve higher insurance coverage as a result of his dedication and never-ending efforts to address challenges in protecting scheme members' benefits. He believes that enforced enrollment has no place in achieving effective insurance coverage.

*"Despite it is not my obligation, I inquire as to why the health center does not provide proper service while we are urging people to enroll. No one ever turns me down for enrollment since they know how enthusiastic I am about it. Anyone who believes I am on his side will put their faith in me. They pay attention to what I have to say. If you support them, they will stand with you. They may be harmed due to events beyond our control, but we must do everything we can to assist them. Why do they pay the premium for the next year if you take the membership fee and remain quiet while they are oppressed and crying?" (KII-18, Kebele leader)*

## Community awareness and acceptability of the CBHI

*Community awareness.* Other scheme performance challenges are emerging from the community's low awareness and unfavorable attitude regarding health insurance. There is low awareness of the type of services they are entitled to receive, the capacity of contracted health centers, the principles underlying CBHI, and the steps they need to follow to secure their benefits, especially for claims reimbursement.

People who are insured have higher expectations of health care services. When their expectations are not met, they tend to blame the system and terminate their membership. Some members of the scheme are also unaware of the notion of solidarity. They wish to receive the benefits of the scheme in exchange for their contributions.

"*Some people claim that 'If I don't get treated, what is the benefit of health insurance?' because the year passes by without treatment. It will, however, benefit them in an emergency. It's due to a lack of understanding.*" (IDI-6, current member)

"*If health insurance is voluntary, no one will renew the membership in our Kebele, except those with known health problems. They have no desire in mutual support, but rather have a desire to be supported. Some people re-enroll after dropping out of health insurance, when they need health care.*" (KII-12, HEW)

*Acceptability of CBHI.* Most of the interviews indicated that health insurance is not well accepted in the community. Certain members of the community did not value the benefits of health insurance. For some of them, CBHI is a political instrument that isn't being used for the community's benefit. They stated that they are paying the premium not to be different from the rest of the population and to avoid confrontation with the *Kebele* leaders. Others are renewing their policy to maintain their relationship with *Kebele* leaders and HEWs who are mobilizing the community. Even some of the insured seek health care in private clinics at their expense, and others pay the premium without renewing their membership identification cards.

"*The community is not interested to enroll in health insurance. We are literally begging people to subscribe or renew their membership. When we visit the community during the renewal time, some residents close their doors and hide. They don't want to hear our voice at that time. They don't want anyone to bring up the topic of health insurance in general.*" (KII-12, HEW)

"*We pay a health insurance membership fee to keep ourselves connected to the rest of the community. Many people pay the membership fees, but never receive the membership card; they do so to avoid being accused by Kebele leaders.*" (IDI-8, previous member)

"*Health insurance is a deception; it simply serves to collect fees and instigate disputes among farmers. The kebele leaders urge us to become member of the health insurance, it seems we have enrolled into something that we did not know exactly.*" (IDI-5, previous member)

*Rumors and misinformation.* Another concern that prevents people from joining or adhering to the scheme is the spread of misleading information in the community. Health care delivery and insurance-related problems are widely publicized; some are exaggerated, while others are rumors or misinformation. When something goes wrong with health care or insurance services, word spreads swiftly throughout the community. Even those who have not encountered the situation appear to be lamenting it as though it has happened to them. In addition, people attempt to portray a distorted image of the community's true problem. Rumors and misinformation concerning health care delivery and insurance services quickly circulate throughout the community, with some emanating from private clinics to attract more consumers. Even an issue that occurred at the time of the introduction of health insurance is still rumored in the community as a fresh problem, despite the fact that it was already resolved.

"*Another is propagation of rumors in the community. If a person faced a problem related to health insurance, he or she can take to the community and spread in a misleading manner.*

*Such rumors are more likely to be heard in society than the stories of people who have benefited from the scheme." (KII-2, health center director).*

"*Rumors and misinformation are widely accepted in the community and are often used to dis-credit health insurance; nevertheless, good things are rarely acknowledged." (KII-1, district health officer)*

Surprisingly, the community, including those who have left out and are criticizing the scheme do not want health insurance to be abolished, since they know they will be able to rejoin and benefit from it at a later point in time.

## Discussion

This study examined the scheme's enrollment trends using key indicators and explored its performance challenges from the perspectives of various stakeholders. The enrollment status at both districts has shown non-linear trends over the study period with both positive and negative growth ratios being noted. In Kallu district, both the coverage and renewal ratios increased sharply between 2017 and 2018, however in Tehulederie, both ratios only slightly changed during the same period. This discrepancy between the two districts might be attributed to the timing of the scheme's initiation, which was two years late in Kallu. Membership growth for a voluntary program is a bit slower at first, and increased as awareness improves over time. Increased growth is expected at the early stage of the scheme because of the lower membership base [12]. The coverage ratio in both districts seems to have stabilized at around 60% since 2018. One plausible explanation is that, in accordance with the bylaw, members won't be permitted to access health care if district-level health insurance coverage falls below 60%. The district administrator who fails to meet the membership requirement will be responsible for the problems caused by the service interruption [40]. As a result, those in charge of membership enrollment may exert extra effort up until this requirement is met. Overall, the scheme in Tehulederie has a relatively higher population coverage and better membership retention. This could be due to the strong foundation laid by a rigorous public awareness campaign throughout the pilot period, as well as the technical assistance provided to the district's relevant stakeholders [34]. As noted from the qualitative interviews, another possible explanation is that the perceived poor quality of health care, particularly the lack of access to medicines in CBHI affiliated health facilities, was a bigger worry in Kallu district than in Tehulederie, which might be impeding the scheme's membership growth capacity.

The low renewal ratios along with an erratic membership growth in Kallu suggests that the scheme is experiencing internal movement, with some households leaving and others joining, potentially exposing it to adverse selection. This will result in a higher claims ratio, lower net income, and maybe bankruptcy if the problems are not fixed [12]. In support of this argument, findings of a companion article of this series on the financial performance of the schemes showed that both schemes experienced an excess claims ratio during 2014 to 2020. The scheme in both districts spent more than it received for claims settlement in almost all the period under the study, and hence experienced heavy losses in these periods. Adverse selection was the key issue that, among other things, led to an increase in the scheme's claims costs [47].

To remain viable, a micro-insurance program must have a minimum growth ratio of zero [11]. However, both schemes have experienced negative growth ratios over the time period under study. Furthermore, population coverage falls short of Ethiopia's Health Sector Transformation Plan, which aspires to 80% coverage by 2020 [48]. It has been noted that this level of insurance coverage was achieved by enforcing different intimidation techniques during the enrollment or renewal periods without the community's discretion, instead of employing

membership attraction or retention strategies. The qualitative data highlighted a number of issues that could be contributing to the observed level of performance in terms of membership enrollment trends. The first and most critical issue impeding membership growth is insurance members' dissatisfaction with quality of care provided by contracted health facilities. Unavailability of medicines, perceived poor quality of medicines, perceived discrimination by health care providers, absence of good interpersonal interactions, and lack of trust in the competency and caring attitude of health care professionals were the main health care quality issues. The common complaints with respect to hospital services include long waiting times, long appointments for non-emergency conditions, and the inability to access emergency care free of charge without a referral letter.

The findings on medicine availability basically corroborate what has been documented in the literature, which has linked it to low enrollment and renewal rates in the scheme. In Ethiopia, the quality of health services, notably the availability of most essential medicines in public facilities, influences household decisions about whether or not to enroll or renew [49]. A study comparing the performance of two districts based on membership enrollment in Tanzania found that the high-performing district had better medicine availability [50]. Overall, scarcity of medicines at contracted service provider facilities was a common problem experienced by insured patients in different settings [28, 51–55]. The perceived low quality of medicines was also identified as a major barrier to insurance subscription. For non-subscribers, a major factor for not to participate in the scheme was the low quality of medicines provided to insurance members [19, 56]. It was also revealed that the insured who were given generic medicines thought the care they received was of inferior quality [57].

Insured households were also dissatisfied with the way health care providers dealt with them. Many felt discriminated against because of their health insurance status, and they expressed their frustration with the disrespectful behavior and uncaring attitude of care providers. This was highlighted during the CBHI pilot phase in Ethiopia, where health care providers did not treat insured patients appropriately, believing that most insured people came to the health facility with minor medical issues due to the free service [28]. For the insured, health care providers were less likely to take their weight and temperature, use a stethoscope, physically examine them, and inform them of their diagnostic results in Burkina Faso [58]. In Rwanda and Ghana, insured patients reported a climate of disrespect and carelessness [17] as well as an unjust queuing system that favors the uninsured [59]. Other studies have also reported that insured patients have perceived and experienced discrimination and verbal abuse from health care providers [18, 19, 23, 51, 52].

It was understood that, due to the increased workload related to insurance coverage, health care providers were unable to display good interpersonal interaction or spend enough time examining and explaining patients' health concerns. It was also apparent that some care providers have shown signs of bitterness as a result of their overworked schedules. This may be partially attributable to the fact that, despite increased workload, there is no incentive system in place for health professionals. However, the claim that insured patients are handled differently than paying patients was mostly based on the insured's perceptions, which seems not really happening. In either scenario, whether there was perceived or actual discrimination, the scheme's ability to attract and retain members would be hampered. Overall, the findings of this study indicated a lack of mutual trust in the relationship between scheme members and health care providers. Insured people associate everything that happens in health facilities with their insurance status, whereas health care providers might be biased in the notion that insured people visit health facilities for every minor ailment owing to free treatment. Improving health care providers' interpersonal communication skills could be crucial to enhance mutual understanding and, as a result, to address most of these concerns.

The findings also revealed that some scheme members had little faith in the overall quality of health care provided by CBHI-affiliated health facilities. This was primarily because of their preferences, expectations, and level of awareness. As a result, people with financial means choose to go to private clinics and pay for their own care. If health care is of poor quality, membership will be less attractive, and expanding membership coverage will be challenging for the relevant stakeholders. It has been documented elsewhere that low perceived quality of care was a major reason for low enrollment and renewal of subscriptions [16, 22].

Despite the fact that quality of care is vital to the success of UHC initiatives, governments have paid little attention to it, which Ridde and Hane (2021) described it as a "known but often ignored challenge" [60]. Quality of care will continue to be a major bottleneck to the scheme's sustainability unless the government devotes significant resources to health system strengthening, particularly human resource development and pharmaceutical supply, in order to meet the rising health care demand. It has been recognized that the scheme was launched without first strengthening the health system's capacity to handle the increasing patient flow and workload without compromising the quality of care, which was the main root cause for the existing problem.

Members of the scheme are entitled to reimbursement for any medical services they bought in private institutions due to a lack of availability in CBHI affiliated health facilities, as long as they follow the appropriate procedures. The scheme's performance in this regard can be assessed using the timeliness of reimbursement and the rejection ratio of submitted claims. The effectiveness with which claims are processed has a direct effect on subscriber satisfaction, which in turn has an impact on membership growth [12]. The findings revealed that claims settlement for OOP expenses was another source of dissatisfaction among insured persons. While OOP spending is a problem in and of itself, clients are not properly reimbursed for their expenditures. Both the delay in reimbursement and the rejection of claims were major complaints. Scheme participants must wait a long time for their claims to be processed, with repeated appointments and long queues at the scheme office. For insured households who reside in remote areas, the cost of traveling to the scheme and the opportunity costs of following up on claims processing outweigh the benefits of receiving the claim. We were unable to get claims reimbursement records from the scheme office in order to determine the claims rejection ratio. However, the qualitative data clearly indicated the depth of the problem. Claims rejection was also common for a number of reasons, most of which stem from clients' low understanding of the kind of paperwork they should submit with their claims. In essence, claims rejection reflects how well insured people understand the scheme [12]. When a claim is denied, it creates a negative perception of the insurance scheme that spreads beyond the claimant, undermining membership enrollment and renewal.

The findings reveal a lack of political will and commitment to the scheme's implementation. Rather than solving the many problems that arise during implementation and striving to serve the needs of the community, the primary concentration is on growing the number of enrollees. The efforts made by higher authorities are once-a-year campaigns that are only apparent at the time of membership enrollment. The scheme will fail to meet its mission as long as respective authorities at all levels of government do not closely monitor and support its implementation. This is corroborated by another study in Nigeria, where non-involvement of local government officials in the implementation of a CBHI scheme negatively affected its effectiveness [61]. The scheme at the district level, where it is operationalized, is not an autonomous sector. Instead, it has been organized under the district health office, with a single sector acting as both a provider and a buyer of health care, raising accountability concerns.

There appear to be some "insurer moral hazard" practices in the scheme's governance, which could be explained by a lack of transparency, the issuance of directives that limit the

insured's benefits, and infringement of governing rules. The insurers have not adequately communicated how the health insurance program works, which may be linked to the scheme's financial strain. Most of the time, scheme members remain in the dark about what they must do in order to receive their claims. This mirrored the findings of the Ethiopian pilot phase evaluation, which found that members went to private pharmacies without knowing what was required for reimbursement, resulting in their claims being rejected and leaving them highly dissatisfied [25].

A new directive was issued at the time of this study, stating that claims reimbursements should be made at the price set by public health facilities. Concerns have been raised about the impact on future membership growth efforts, as well as the risk of jeopardizing scheme membership benefits. It will threaten both access to health care and the scheme's capacity to provide financial protection. Although the purpose of health insurance is to eliminate uncertainty, the amount of money that will be reimbursed will be unpredictable as a result of this directive. The poor will be hit harder, as it is likely that a large portion of the bill will go unpaid, or that they will forego treatment because they are unsure how much of their OOP expenses will be reimbursed. Instead of limiting claims benefits, it is essential to consider certain limits on the number of visits in a given period to discourage unnecessary visits to health facilities. Infringement of governing rules, which results in legitimate benefits being withheld from members, is another form of governance concern that could hamper membership growth.

The fact that some households pay the premium without renewing their membership implies that the scheme is undervalued, and that scheme administrators are equating insurance coverage with revenue collection regardless of whether or not scheme members use health services and being protected from financial hardship. It also signifies that enrollment is compulsory. To be successful, key stakeholders involved in membership mobilization should focus on protecting the interests of the community. Building mutual trust with the community should be a priority, according to best practices learned from a high-performing village. Those who have direct contact with the community during membership enrollment, such as *Kebele* leaders and HEWs, should demonstrate commitment to assisting scheme members in addressing their complaints. People's resistance to membership enrollment will be substantially reduced if they become community advocates. It has been argued that in order for a CBHI to be effective in terms of population coverage, voluntarism should be abandoned. Mandatory enrollment, on the other hand, can only work if governments commit to considerably expanding public funding of the health system and subsidizing members' contributions [14]. People may have a feeling of ownership if enrollment is voluntary, which may improve community members' participation in decision-making. Due to the existing weak financial systems and poor quality of health care, mandatory enrollment will cause the community to develop a negative attitude towards the scheme. Furthermore, if people are forced to participate in the scheme, they may engage in abusive behaviors.

Another concern that prevents people from joining or adhering to the scheme is the spread of misleading information in the community. Problems are widely publicized; some are overstated, while others are rumors or misinformation. This will severely compromise the sustainability of the scheme unless they are proven to be spurious claims. It contributes to the development of negative attitudes regarding health insurance in the community. The community does not value the scheme, and for some it has been seen as a political tool that is not being used for the community's benefit. The majority of people only want to join health insurance when they are facing a serious illness, and quit the scheme when they no longer need care. This necessitates the implementation of awareness raising strategies that could help in the debunking of misinformation and rumors surfacing in the community.

The study's main limitation is that it was unable to investigate the overall scheme execution period due to insufficient data. Furthermore, due to a lack of records on rejected claims, we were unable to determine the claims rejection rate for the schemes. The use of secondary data may not accurately reflect real enrollment trends. The findings will, however, provide valuable insight for the study districts, as well as other areas with similar setups that would like to initiate improvements. It will be an essential input for policymakers as they strive to establish higher-level pools and revise scheme designs.

## Conclusions

The scheme experienced negative growth ratios in both study sites over the study period, showing that the program is not functionally viable. The low quality of care was a key barrier to membership development. It will remain a major bottleneck to the scheme's sustainability unless the government invests significantly in health system strengthening, particularly human resource development and pharmaceutical supply, in order to meet growing health care demand. The delay or rejection of claims reimbursement for insurance holders was another threat to membership growth. The presence of certain "insurer moral hazard" tactics, which seek to reduce the scheme's financial burden, perpetuates this problem. All these will erode the community's trust in the scheme, which in turn will have an impact on membership growth. Compulsory enrollment is an implicit approach for increasing membership coverage, but it is unlikely to be a long-term solution unless the government focuses on quality improvement initiatives. Instead, scheme administrators and other stakeholders should foster a trusting climate among community members and develop less costly attraction strategies. It is suggested that awareness raising strategies be implemented to help debunk misconceptions and rumors that are prevalent in the community. Overall, the scheme will inevitably fail to attain its mission unless relevant stakeholders at all levels of government demonstrate political will and commitment to its implementation, as well as advocate for the community.

## Acknowledgments

We would like to express our deepest gratitude to the health offices of Tehulederie and Kallu districts, health extension workers, and *kebele* leaders. I (MH) want to acknowledge Bahir Dar university for the opportunity it has given me to pursue my PhD study.

## Author Contributions

**Conceptualization:** Mohammed Hussien.

**Data curation:** Mohammed Hussien, Muluken Azage, Negalign Berhanu Bayou.

**Formal analysis:** Mohammed Hussien, Muluken Azage, Negalign Berhanu Bayou.

**Investigation:** Mohammed Hussien, Muluken Azage, Negalign Berhanu Bayou.

**Methodology:** Mohammed Hussien, Muluken Azage, Negalign Berhanu Bayou.

**Project administration:** Mohammed Hussien, Muluken Azage, Negalign Berhanu Bayou.

**Resources:** Mohammed Hussien, Muluken Azage, Negalign Berhanu Bayou.

**Supervision:** Muluken Azage, Negalign Berhanu Bayou.

**Validation:** Mohammed Hussien, Muluken Azage, Negalign Berhanu Bayou.

**Visualization:** Mohammed Hussien, Muluken Azage, Negalign Berhanu Bayou.

**Writing – original draft:** Mohammed Hussien.

**Writing – review & editing:** Mohammed Hussien, Muluken Azage, Negalign Berhanu Bayou.

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
