## [Decision Letter · Decision Letter 0]

29 Jul 2022

PONE-D-22-08512A mixed methods study of community-based health insurance enrollment trends and underlying challenges in two districts of northeast Ethiopia: a proxy for its sustainabilityPLOS ONE

Dear Dr. Hussien,

Thank you for submitting your manuscript to PLOS ONE. After careful consideration, we feel that it has merit but does not fully meet PLOS ONE’s publication criteria as it currently stands. Therefore, we invite you to submit a revised version of the manuscript that addresses the points raised during the review process.

We look forward to receiving your revised manuscript.

Kind regards,

Pablo Andrés Villalobos Dintrans, DrPH

Academic Editor

PLOS ONE

Journal Requirements:

Additional Editor Comments:

Dear Dr. Hussein

Besides the reviewer's comments I think the article needs to better establish the way in which quantitative and qualitative information was used. So far, it seems you just presented some data (context information) and then most results are based on the interview. Currently, the article fails to show why this is a "mixed methods" study. Please expand the Methods section to explain how both pieces of information interacted to get the results.

Reviewers' comments:

Reviewer's Responses to Questions

**Comments to the Author**

1. Is the manuscript technically sound, and do the data support the conclusions?

Reviewer #1: Yes

2. Has the statistical analysis been performed appropriately and rigorously? 

Reviewer #1: Yes

3. Have the authors made all data underlying the findings in their manuscript fully available?

Reviewer #1: Yes

4. Is the manuscript presented in an intelligible fashion and written in standard English?

Reviewer #1: Yes

5. Review Comments to the Author

Reviewer #1: Manuscript: A mixed methods study of community-based health insurance enrollment trends and underlying challenges in two districts of northeast Ethiopia: a proxy for its sustainability

The manuscript addresses a very relevant and interesting topic for health systems, especially in low- and middle-income settings. However, it will benefit from a revision of the following aspects:

• Abstract

Background: maybe adding a brief definition of community-based health insurance

Methods: Why were these two districts selected or included in the study?

For quantitative data, it is relevant to indicate the period of analysis (2017-2021).

Results: What does “inconsistent trends” mean? What does “better enrollment trend” mean?

Discussion: How are the inconsistent trends related to not functional viability?

Introduction: before indicating where the CBHI have been adopted, it would be beneficial to briefly explain what type of arrangement the CBHI is. Why is it appropriate for covering subsistence farmers and informal sector employees?

Lines 91-95. Is it the unified pool system voluntary or mandatory? Universal or just for formal workers? It seems that those characteristics would determine if the findings regarding CBHI are more (or less) relevant for its implementation. Also, if the system is not already defined, the study certainly will provide insights for its design.

Materials and methods

Lines 114-123. It is not clear if the schemes sources of revenue and benefit packages detailed in this paragraph apply to all CBHI or there is some level of variation between districts.

Lines 131 and 132. As it is the first time the abbreviation HEWs is mentioned, it should be explained here.

Lines 150-154. How are the 19 key informant interviews and the in 9 depth interviews distributed between the two districts? As they are treated as two different study case it might be relevant o have the disaggregated information.

Lines 160-161. What is the total period of implementation of the scheme (2011 and 2013, depending on the district?)?

Line 162. What are the informal field interviews and why are not considered in the study design and participant selection (as the KIIs and IDIs are)?

Line 181. How is the target population (see line 6) defined to calculate the coverage ratio?

Line 182. How is defined the “eligible” population for renewal (see line 64), in order to calculate the renewal ratio?

Results

Lines 213-216. Use commas as separator for numbers (consistency).

Lines 214-215. How were the target populations estimated (is it the total population for each of the districts)?

Table 1, Figure 1. Is there really a trend or pattern in the data displayed? Results seem different for the two districts, is it possible from the qualitative analysis to further explain why?

Lines 230-231. The enrollment trend is based on the coverage ratio indicator?

Lines 261. Is it OK medicines prescribed, or it refers to medicines dispensed/provided/delivered at health centers? Not sure if it is completely clear if the medicines are delivered free of charge at public centers, or if member should claim reimbursement after paying for their medicines at these facilities.

Lines 330-340. Maybe to consider in the study setting section. Why is it possible for the two districts to have different claims reimbursement systems? Is it a district level decision?

Line 430. Correct “comminity’s”.

Discussion

Lines 529-532. The characterization of the trend is not precise. Terms as inconsistent or better trend are used but not necessarily explained. Also according to Table 1 Tehulederie district shows higher coverage and renewal during the period, but Kallu’s coverage and renewal show higher increases (moving from 27.2 to 58.5 and 37 to 62.6 respectively).

Lines 537-538. Is it possible to include in the quantitative analysis indicators such as the ones mentioned here (claims ratio and net income) at least for one of the districts or for shorter period? It would be an interesting adding to the analysis.

Line 579. Is it possible state (form the collected data) that the described/perceived situation is not really happening?

Line 583. Is it possible to state (from the collected data) that the providers perception is biased?

General comments:

It is hard to find the ex-members voice as there are not quotations from them (or just one), and their perceptions are not necessarily individualized in the analysis.

The two components of the mixed methods seem a bit unbalanced, as the quantitative analysis is simpler than the qualitative analysis, and it is not that clear how the results from the former are considered.

Health care providers attitude, behavior and competency. I know these is based on perceptions from the different stakeholders, but are there any reasons behind this behavior? Does CBHI underpay health centers for the services?

Claims reimbursement for insurance holders. Not sure if it is completely clear. Are all the services provided in public and private facilities subject to reimbursement claims? Are there some services free of charge at the point of delivery?

What are the conditions for the uninsured? Do they pay much more? Or, are they healthier (due to adverse selection)? Is there something that can be said about them?

6. PLOS authors have the option to publish the peer review history of their article (what does this mean?). If published, this will include your full peer review and any attached files.

Reviewer #1: No

---

## [Author Response · Author response to Decision Letter 0]

1 Aug 2022

Editor Comments:

1. Besides the reviewer's comments I think the article needs to better establish the way in which quantitative and qualitative information was used. So far, it seems you just presented some data (context information) and then most results are based on the interview. Currently, the article fails to show why this is a "mixed methods" study. Please expand the Methods section to explain how both pieces of information interacted to get the results.

--- Thank you very much for the comment. In a mixed methods study the two components may have equal weight, or it may be primarily quantitative or qualitatively driven. Our study primarily aims to explore the challenges for membership development, we seek to include the quantitative data to understand the enrollment status and hence to complement the findings with the qualitative interviews, believing that it will add value to the study. We tried to show how the two components are used and integrated under “study design” [line 180-195]. The two components are integrated during data interpretation. The quantitative components discussed first and the qualitative themes are discussed and linked to the quantitative data (how they contribute the observed level of scheme performance in terms of membership growth). 

Reviewer #1 comments: 

The manuscript addresses a very relevant and interesting topic for health systems, especially in low- and middle-income settings. However, it will benefit from a revision of the following aspects:

Abstract:

1. Background: maybe adding a brief definition of community-based health insurance

--- Thanks. Revised accordingly [line 21-23]

2. Methods: Why were these two districts selected or included in the study?

For quantitative data, it is relevant to indicate the period of analysis (2017-2021).

--- Revised as to the recommendation. …purposively selected districts [line 31], The purpose is detailed under participant selection [line 197-201]

--- Data were collected from 2017 to 2021 [line 33]

3. Results: What does “inconsistent trends” mean? What does “better enrollment trend” mean? 

--- These terms are replaced by other terms which we consider are appropriate [line 42-46]

4. Discussion: How are the inconsistent trends related to not functional viability?

--- Thank you for your critical look. It is the negative growth ratios that make the schemes functionally not viable. We revise it accordingly [line 53-54]

Introduction: 

5. Before indicating where the CBHI have been adopted, it would be beneficial to briefly explain what type of arrangement the CBHI is. Why is it appropriate for covering subsistence farmers and informal sector employees?

--- Thank you for your kind advice . We revise it according to the suggestion [line 66-78]

6. Lines 91-95. Is it the unified pool system voluntary or mandatory? Universal or just for formal workers? It seems that those characteristics would determine if the findings regarding CBHI are more (or less) relevant for its implementation. Also, if the system is not already defined, the study certainly will provide insights for its design.

--- The unified pool system is yet to be determined. It is the future intention as indicated in the health financing strategy. We include terms showing that it is at a plan stage (aims to…line 120, the proposed …line 123)

Materials and methods:

7. Lines 114-123. It is not clear if the schemes sources of revenue and benefit packages detailed in this paragraph apply to all CBHI or there is some level of variation between districts.

--- Thank you again for the question. It applies to all CBHI. To avoid this unclarity, we describe (shift ) the characteristics of CBHI in Ethiopia - to the introduction section with a sub-title “A brief overview of CBHI in Ethiopia” 

8. Lines 131 and 132. As it is the first time the abbreviation HEWs is mentioned, it should be explained here.

--- revised accordingly [line 176-177]

9. Lines 150-154. How are the 19 key informant interviews and the in 9 depth interviews distributed between the two districts? As they are treated as two different study case it might be relevant to have the disaggregated information.

--- Thanks again. The initial plan was to take equal number of participants from each district. Later sample was determined based on data saturation. Disaggregated information is provided, but not specific to each participant type [Line 211 and 215]

10. Lines 160-161. What is the total period of implementation of the scheme (2011 and 2013, depending on the district?)?

--- It is 10 years in Tehulederie since 2011 and seven years in Kallu since 2013 and now mentioned {line 230]

11. Line 162. What are the informal field interviews and why are not considered in the study design and participant selection (as the KIIs and IDIs are)?

--- Thanks for your comment. Included under participant selection [221-223], and defined what does it mean [241-243]

12. Line 181. How is the target population (see line 62) defined to calculate the coverage ratio?

--- Both target population and “eligible” for renewal are defined under data analysis[line 258-262]. The definition of coverage, growth and renewal ratios are also provided under data analysis and removed from the introduction part to avoid duplication.

13. Line 182. How is defined the “eligible” population for renewal (see line 64), in order to calculate the renewal ratio?

--- addressed above [#13)

Results:

14. Lines 213-216. Use commas as separator for numbers (consistency).

--- Edited accordingly [line 295-298]

15. Lines 214-215. How were the target populations estimated (is it the total population for each of the districts)?

--- The definition is provided under the data analysis (Line 258-260)

16. Table 1, Figure 1. Is there really a trend or pattern in the data displayed? Results seem different for the two districts, is it possible from the qualitative analysis to further explain why?

--- Thank you again for raising this issue. A trend is a general direction in which something is changing. A pattern is a set of data that follows a recognizable form, a repeated occurrence or sequence. Accordingly Figure 1 shows a trend. Results seem different for the two districts. We could not explain why this is so from the qualitative analysis. But we provide further possible explanations [line 637 - 650] 

17. Lines 230-231. The enrollment trend is based on the coverage ratio indicator?

--- Thank you for your kind observation. It is not only based on the coverage ratio, now we added the renewal ratio to the figure (Fig 1)

18. Lines 261. Is it OK medicines prescribed, or it refers to medicines dispensed/provided/delivered at health centers? Not sure if it is completely clear if the medicines are delivered free of charge at public centers, or if member should claim reimbursement after paying for their medicines at these facilities.

--- Dispensed/provided might be equally or more appropriate, so revised accordingly [line 345]. As long as the medicines are available in the contracted health facilities, they receive it free of charge. We make it clear this [line 153-157]

19. Lines 330-340. Maybe to consider in the study setting section. Why is it possible for the two districts to have different claims reimbursement systems? Is it a district level decision?

--- This decision is not stated in the directive. The decision was made at district level to facilitate the reimbursement process and to decrease additional costs for members. Rather than taking it as a study setting, we opt to consider this as part of the study finding. Further elaboration is provided how it is practiced between the two districts [434-438] 

20. Line 430. Correct “comminity’s”.

--- Edited as community’s [line 528]

Discussion:

21. Lines 529-532. The characterization of the trend is not precise. Terms as inconsistent or better trend are used but not necessarily explained. Also, according to Table 1 Tehulederie district shows higher coverage and renewal during the period, but Kallu’s coverage and renewal show higher increases (moving from 27.2 to 58.5 and 37 to 62.6 respectively).

--- Thanks again. We tried to characterize the trend differently based on the suggestion [line 636 and 650]. The difference among the two districts with respect to Table 1 (the new Fig 1) is given more explanations [line 637 - 650]

22. Lines 537-538. Is it possible to include in the quantitative analysis indicators such as the ones mentioned here (claims ratio and net income) at least for one of the districts or for shorter period? It would be an interesting adding to the analysis.

--- Thank you for your critical look. It is possible. We have included findings on these indicators, which is based on findings of a companion article of this series on the financial viability of the scheme [line 657-662]

23. Line 579. Is it possible state (form the collected data) that the described/perceived situation is not really happening?

--- Yes of course. What we understand from the key informant interviews is that at the time of the study, discrimination on the base of insurance status is not really happening. Even some stated that health care providers favor the scheme members, since membership attraction is partly their responsibility. But it is better not to conclude at all. So revised as ... which seems not really happening [line 709]

24. Line 583. Is it possible to state (from the collected data) that the providers perception is biased?

--- Thanks again. Not sure, but may be biased. Hence, we revise it as “health care providers might be biased…” [line 714]

General comments:

25. It is hard to find the ex-members voice as there are not quotations from them (or just one), and their perceptions are not necessarily individualized in the analysis.

--- Thank you for this constructive feedback. we did not look at who was represented, rather we only consider what was said and take the idea that most strikes us. Now we try to include their voice, and it should be [line 365, 383, 608].

26. The two components of the mixed methods seem a bit unbalanced, as the quantitative analysis is simpler than the qualitative analysis, and it is not that clear how the results from the former are considered.

--- Thank you for the comment. This is also the concern of the editor. Yes of course the two components are unbalanced, which is one characteristics of a mixed methods study. They may have equal weight, or primarily quantitative or qualitatively driven. Our study primarily aims to explore the challenges for membership development, but we seek to include the quantitative data to understand the enrollment status and hence to complement the findings with the qualitative interviews, believing that it will add value to the study. We tried to show how the two components are used and integrated under “study design” [line 180-195]. The two components are integrated during data interpretation. The quantitative components discussed first and the qualitative themes are discussed and linked to the quantitative data (how they contribute the observed level of scheme performance in terms of membership growth). 

27. Health care providers attitude, behavior and competency. I know these is based on perceptions from the different stakeholders, but are there any reasons behind this behavior? Does CBHI underpay health centers for the services?

--- As stated in the manuscript, Health care providers may not demonstrate excellent interpersonal communication or spend enough time examining and discussing patients' health concerns due to increased workload associated with insurance coverage. Furthermore, despite the burden, there is no incentive system in place for healthcare professionals (now we include the later as one possible reason - line 706-708). CBHI pays health centers for the services, and the problem is a delay in claims payment. But this is not related to health care professionals’ behavior as this does not affect their personal benefit.

28. Claims reimbursement for insurance holders. Not sure if it is completely clear. Are all the services provided in public and private facilities subject to reimbursement claims? Are there some services free of charge at the point of delivery?

--- This is well clarified now [line 153-157] 

29. What are the conditions for the uninsured? Do they pay much more? Or, are they healthier (due to adverse selection)? Is there something that can be said about them?

--- Low membership coverage indicated the inability to attract healthy individuals leading to adverse selection. This paper aims to explore the reasons for low membership development. Issues related to adverse selection have been addressed in companion article of this series. Now, we state adverse selection as one reason for high claims cost (line 661]

We are grateful for your constructive feedback!

---

## [Editor Report · Decision Letter 1]

4 Aug 2022

PONE-D-22-08512R1A mixed methods study of community-based health insurance enrollment trends and underlying challenges in two districts of northeast Ethiopia: a proxy for its sustainabilityPLOS ONE

Dear Dr. Hussien,

Thank you for submitting your manuscript to PLOS ONE. After careful consideration, we feel that it has merit but does not fully meet PLOS ONE’s publication criteria as it currently stands. Therefore, we invite you to submit a revised version of the manuscript that addresses the points raised during the review process.

We look forward to receiving your revised manuscript.

Kind regards,

Pablo Andrés Villalobos Dintrans, DrPH

Academic Editor

PLOS ONE

Journal Requirements:

Additional Editor Comments (if provided):

I think the authors did a good job addressing the comments. I still would like a more sound discussion on the methods and the way this study can be viewed as "mixed methods". These references can be helpful for better defined what is a mixed method study, what type was used, and why it makes sense for this particular study and its aim:

Shorten A, Smith J. 

Mixed methods research: expanding the evidence base. 

Evidence-Based Nursing 2017;20:74-75.

Abdi-Rizak M. Warfa. Mixed-Methods Design in Biology Education Research: Approach and Uses. CBE—Life Sciences Education 2016 15:4

Also, I think you need a better response to comment 16 (results) [16. Table 1, Figure 1. Is there really a trend or pattern in the data displayed? Results seem different for the two districts, is it possible from the qualitative analysis to further explain why?]. This is exactly what I would expect to see the advantage of using a mixed method study, in which you explain some of the observed (quant) results using your methodology (qual). Please expand this discussion further.
---

## [Author Response · Author response to Decision Letter 1]

6 Aug 2022

Editor Comments:

1. I think the authors did a good job addressing the comments. I still would like a more sound discussion on the methods and the way this study can be viewed as "mixed methods". These references can be helpful for better defined what is a mixed method study, what type was used, and why it makes sense for this particular study and its aim:

--- Thank you very much for the comment. We thank also for the materials you shared with us. We tried to made some revisions on the methods section as to the comments [Line 176-187] - Manuscript with Track Changes

2. Also, I think you need a better response to comment 16 (results) [16. Table 1, Figure 1. Is there really a trend or pattern in the data displayed? Results seem different for the two districts, is it possible from the qualitative analysis to further explain why?]. This is exactly what I would expect to see the advantage of using a mixed method study, in which you explain some of the observed (quant) results using your methodology (qual). Please expand this discussion further.

--- Thank you again for raising this issue. In addition to what we have do during the first revision, we try to expand the discussion further [Line 642-646] - Manuscript with Track Changes

---

## [Editor Report · Decision Letter 2]

11 Aug 2022

A mixed methods study of community-based health insurance enrollment trends and underlying challenges in two districts of northeast Ethiopia: a proxy for its sustainability

PONE-D-22-08512R2

Dear Dr. Hussien,

We’re pleased to inform you that your manuscript has been judged scientifically suitable for publication and will be formally accepted for publication once it meets all outstanding technical requirements.

Thanks for your patiente and responsiveness to the comments, and congratulations.

Kind regards,

Pablo Andrés Villalobos Dintrans, DrPH

Academic Editor

PLOS ONE
---

## [Editor Report · Acceptance letter]

19 Aug 2022

PONE-D-22-08512R2 

A mixed methods study of community-based health insurance enrollment trends and underlying challenges in two districts of northeast Ethiopia: a proxy for its sustainability 

Dear Dr. Hussien:

I'm pleased to inform you that your manuscript has been deemed suitable for publication in PLOS ONE. Congratulations! Your manuscript is now with our production department. 

Kind regards, 

on behalf of

Dr. Pablo Andrés Villalobos Dintrans 

Academic Editor

PLOS ONE